# Reported Cases of Alcohol Consumption and Poisoning for the Years 2015 to 2022 in Hail, Saudi Arabia

**DOI:** 10.3390/ijerph192215291

**Published:** 2022-11-19

**Authors:** Taghreed Alhaidan, Abdullah R. Alzahrani, Abdulwahab Alamri, Abrar A. Katpa, Asma Halabi, Alaa H. Felemban, Safaa M. Alsanosi, Saeed S. Al-Ghamdi, Nahla Ayoub

**Affiliations:** 1Department of Pharmacology and Toxicology, Faculty of Medicine, Umm Al-Qura University (UQU), Makkah 24375, Saudi Arabia; 2Saudi Toxicology Society, Umm Al-Qura University (UQU), Makkah 24375, Saudi Arabia; 3Department of Pharmacology and Toxicology, Faculty of Pharmacy, University of Hail, Hail 55255, Saudi Arabia; 4Department of Nephrology, King Khalid Hospital, Hail 55421, Saudi Arabia; 5Department of Pharmacy, Makkah Healthcare Cluster, Primary Health Care, Alhusainiah, Makkah 24249, Saudi Arabia

**Keywords:** alcohol consumption, Saudi Arabia, health impact, alcohol poisoning, social influences

## Abstract

This study aimed to determine the pattern of alcohol consumption and its poisoning among the Saudi population in the city of Hail, KSA. Data from a retrospective cohort were collected qualitatively at King Khalid Hospital (KKH) and Hail General Hospital (HGH), covering 550 participants from 2015 to 2022. Two groups were formed comprising patients admitted to the emergency room (ER) and community members; their ages ranged from 19 to 75 years. Group 1 contained 400 participants, of which 250 were patients (244 males, six females) who came to the (ER) with a suspected alcohol overdose or poisoning, and 150 were patients (128 males and 22 females) who were discharged from the (ER) with minimal complaints because of their drinking. Group 2 comprised 150 participants (128 males, 22 females) who were community members, who were surveyed using a questionnaire or interview. In Group 1, 30% of patients reported an altered state of consciousness as a major complaint, 28.8% of patients exhibited abnormal liver function tests (LFTs), 27% had abnormal renal function tests (RFT) with decreased glomerular filtration rates (GFR) and elevated levels of urea and creatinine or low levels of electrolytes or calcium, and 35.6% patients showed elevated levels of pancreatic enzymes. One death was reported due to high alcohol consumption. In Group 2, the community participants reported that they started drinking alcohol due to the influence of other people (29%), stress (11%), depression (10.8%), curiosity (4.4%), and boredom (4%). In addition, 77% of participants were frequent alcohol drinkers and 20% consumed it daily. Further, 68.7% claimed to drink alcohol for more than one hour at a time, while 83.3% experienced blackouts and 70% had problems related to their liver. Moreover, 72.7% of the participants ended up in the hospital and 34.6% suffered from multiple chronic diseases. It is concluded that social influences and stress contributed to the initiation of alcohol use. Despite data gaps, the findings of this study provide a practical understanding of alcohol consumption among the Saudi population and guidance for policymakers.

## 1. Introduction

Alcohol is widely consumed all over the world as a constituent of traditional or leisure activities [1]. Its use differs among countries and ethnicities. In most western cultures, drinking alcohol is a common practice, especially on ceremonial occasions [2]. In contrast, its use is less common in Arabic countries and it is even prohibited in some of them, including Saudi Arabia [3]. Drinking alcohol is not easily accepted in Arabic–Islamic countries because of the socio-cultural and religious values of Muslim societies [4].

Regardless of social, cultural, legal, and religious constraints in Saudi Arabia against the use of alcohol, drinking and alcohol abuse still exist [5]. Besides Saudi Arabia, alcohol consumption has been prevalent in many Arabic and Muslim countries, including Jordan [3], Egypt [6], and Pakistan [7]. Alcohol consumption involves diverse behaviors and has numerous determinants. Many risk factors lead to the misuse of alcohol and dependence and its resultant long-term effects [8]. Generally, males drink more alcohol than females around the world, and females from developed countries are more inclined toward alcohol consumption than those in developing countries [9].

Various factors lead to alcohol drinking and abuse, including personality traits, negative and positive experiences in life, peer influences, and social and parental influences [10]. Social influences and stress contribute significantly to the initiation of alcohol use [1]. For instance, studies found a positive association between the digital marketing of alcohol and greater use of alcohol and even binging or risky drinking activities. Moreover, the marketing of alcohol on digital media has increased in recent years and is reaching more people than the typical broadcast platforms [11,12].

Alcohol is linked to a wide range of health problems, such as acute intoxication, memory impairment, dependence syndrome, alcoholic hepatitis, and alcohol-induced pancreatitis [13,14]. The present study aimed to determine the patterns of alcohol consumption and its poisoning among the Saudi population in the city of Hail, Saudi Arabia.

## 2. Materials and Methods

### 2.1. Study Design

A qualitative methodology was selected for this study, which was conducted through interviews or questionnaires and using data from the medical records of the study population. The plan was to elucidate individuals’ reasoning regarding alcohol consumption and its impact on their health, to inform potential targeted avoidance measures. The study was conducted at King Khalid Hospital (KKH) and Hail General Hospital (HGH) in Hail, Saudi Arabia. The study design was a retrospective cohort design, conducted between June 2015 and June 2022.

### 2.2. Participants

The participants were a combination of patients admitted to the hospital and community members. A total of 550 cases were noted with suspicion of alcohol-associated problems. The participants were divided into two groups. Group 1 comprised 400 patients with symptoms who were admitted to the (ER). Of these 400 participants, 150 patients were discharged from the ER because of the minimal nature of their complaints, and 250 patients were evaluated for alcohol abuse and poisoning through their medical records. Group 2 consisted of 150 community members who were surveyed using a questionnaire or interviews that contained the same questions. Interviews were applied for participants who could not speak English.

### 2.3. Data Collection

Questionnaires and interviews to collect the data were conducted in Hail, Saudi Arabia. The questions were open-ended and were chosen to obtain comprehensive information about the topic under discussion. Demographic data on factors including gender, age, marital status, nationality, and education level were collected. Data on each participant’s initial encounters with alcohol, drinking habits over time, frequency of drinking, and the ill effects of alcohol on their health were gathered. Purposive sampling was used to collect information from both genders and to allow the analysis of a wide range of drinking habits.

### 2.4. Biochemical Parameters

Liver function tests: Normal serum total bilirubin varies from 2 to 21 μmol/L, and a value above 24 μmol/L indicates abnormal liver tests. Normal serum levels of alanine amino transferase (ALT) and aspartate amino transferase (AST) are 7–56 U/L and 8 to 48 U/L, respectively, while elevated values up to 300 U/L in ALT and 81 U/L in AST are abnormal in liver tests. The normal level of gamma glutamyl transferase (GGT) is 9 to 85 U/L, and elevated serum GGT levels of more than 10-times higher are recorded as alcoholism.

Renal function tests: Urine albumin-to-creatinine ratio (UACR) 30 or above is conserved an abnormal value. The reference range for serum creatinine is 0.7–1.2 (mg/dL) for males and 0.5–1.0 mg/dL for females. Levels higher than 1.2 mg/dL for males or 1.0 mg/dL for females are considered abnormal.

Pancreatic function test: The reference lipase range for adults under age 60 is 10–140 U/L. For adults over 60, the normal range is 24–151 U/L. If a patient’s lipase levels are 3–10 times the reference value, this can indicate an abnormal value.

### 2.5. Data Analysis

Interviews were audio-recorded with permission, transcribed verbatim, and saved into NVivo software. Questionnaire and interview data (Appendix A) were analyzed using SPSS version 16.

### 2.6. Ethical Approval

Ethical approval was received from the Ethics Committee, namely the Institutional Review Board (IRB) Committee, Research and Study Department, Hail, Saudi Arabia, No. 2021/18. Informed written consent was obtained from each participant, and, in line with the terms agreed by the participants, their personal data were kept confidential.

## 3. Results

### 3.1. Group 1

A total of 250 patients out of 400 participants were enrolled in Group 1 and assessed for alcohol abuse or poisoning. Group 1 had 244 males and six females, and the average age was 28 (±9.2) years Table 1. They were mostly Saudi nationals (241). The group had 193 single participants and 55 married participants, and the marital status of two was unknown. Participants were also classified on the basis of alcohol poisoning, where the highest number of poisoning cases was noted in the group aged 19 to 30 years, with 167 (77%) cases, followed by 31 to 59 years, with 81 (32%) cases, and 60 to 75 years, with only two participants.

Of the patients who had severe complaints, 60 (30%) had an altered state of consciousness, 50 (25%) experienced vomiting, 40 (20%) experienced confusion, 30 (15%) had jaundice, and 20 (10%) had abdominal pain. Most patients with positive alcohol consumption presented with complaints of an altered level of consciousness, vomiting, confusion, and jaundice; see Figure 1.

The patients underwent liver function tests (LFTs), and 72 (28.8%) exhibited abnormal liver function; see Figure 2. The remaining 178 (71.2%) patients were found to have normal liver function. It was found that 67 patients had abnormal renal function test (RFT) results, while the rest (183) had normal RFT results, as shown in Figure 3. The pancreatic functions of more than one third of the patients were found to be abnormal, i.e., 89 (35.6%), as shown in Figure 4. One death was reported due to alcohol abuse during the study period. This patient was unconscious upon arriving at the ER.

### 3.2. Group 2

In Group 2, 150 participants from the community who had a history of alcohol consumption were interviewed or responded to the questionnaire. Table 2 describes the demographic profile of the Group 2 participants. The mean age of the participants was 31 (±10) years, and all were Saudi nationals. They were also divided according to the driving force that caused them to begin consuming alcohol at different ages, where 44 (30%) were aged 19 to 30 years, 100 (66%) were aged 31 to 59 years, and 6 (4%) were aged 60 to 75 years, and they all had different educational levels.

Participants often mentioned that their first interactions with alcohol had occurred several years ago. The lengths of time for which the participants reported consuming alcohol included 44, 35, 10, 8, 6, 3, and 2 years. Most participants in the study began drinking two years previously. Reasons provided by the participants for their consumption of alcohol are shown in Figure 5.

The data show that the main reason for alcohol consumption was the influence of friends, relatives, or other people. A total of 74 participants (29%) reported that they started drinking alcohol due to the influence of other people. The other reasons reported were as follows: stress (28; 11%), depression (27; 10.8%), curiosity (11; 4.4%), and boredom (10; 4%). Figure 6 shows that 115 participants (76.7%) stated that they frequently drank alcohol and that 35 participants (23.3%) said that they did not drink it frequently.

Furthermore, the data show that most participants consumed alcohol two to four times a week (36%) or on special occasions (24%), while 20% consumed alcohol daily and 20% consumed it weekly (Figure 7).

As shown in Figure 8, most participants (68.7%) claimed that when they were drinking, they consumed alcohol for more than one hour. Some participants (29; 19.3%) drank alcohol for 30 min at a time, and 18 participants (12%) drank it for an hour at a time. When participants were asked if they had ever passed out, blacked out, or experienced memory loss because of drinking, 125 individuals (83.3%) said that they had passed out. Only 25 participants (16.7%) had not experienced blacking out.

A total of 105 participants (70%) responded ‘yes’ to the question of whether they had experienced problems related to their liver or pancreas due to alcohol consumption. The remaining 45 participants (30%) denied experiencing any such liver- or pancreas-related health problems. In addition to blackouts in Figure 9 and liver- or pancreas-related problems, participants were asked about any deterioration in their health after drinking alcohol. In response, 104 participants (69.3%) said that their health had deteriorated because of alcohol consumption, while 46 participants (30.7%) said that their health had not worsened. The participants were asked whether they had ever ended up in hospital because of alcohol intake. It was found that 109 participants (72.7%) had ended up in the hospital due to alcohol consumption and 41 participants (27.3%) had not, as shown in Figure 10.

Chronic diseases were reported as follows: around one third of the participants (48; 32%) had not developed any chronic disease, while 52 participants had multiple chronic diseases due to alcohol consumption, as shown in Figure 11. The most-reported diseases were related to the pancreas and liver.

The effects of alcohol consumption were reported by the participants, with the exception of 12 participants who experienced no ill effects. These effects were related to their social, personal, and psychological lives and their health. The majority shared that their general and mental health had deteriorated, while a few reported that they had lost their jobs and their family lives.

## 4. Discussion

Alcohol poisoning is one of the most frequently encountered medical emergencies worldwide and is associated with high disease and death rates [15]. Knowledge about the prevalence of alcohol use and abuse is important to control this issue. Saudi Arabia is not immune from these issues: alcohol-related toxicity still constitutes one of the main health hazards in Saudi Arabia [16,17]. Unfortunately, an inadequate quantity and quality of data are available on alcohol prevalence and toxicity in Saudi Arabia.

The current study took place at KKH and HGH in Hail, Saudi Arabia, over seven years, from January 2015 to February 2022. The study aimed to determine the pattern of alcohol consumption and its toxicity among the Saudi population in Hail City, Saudi Arabia. The goal was to produce findings that contribute to an up-to-date understanding of alcohol use among the Saudi population and its implications for their health. Despite specific personality traits associated with the earlier consumption of alcohol and more frequent and heavy drinking, findings along these lines were not consistent in this study. The main reason for alcohol consumption among the participants was the influence of other people, followed by stress, depression, boredom, and curiosity.

The drinking patterns observed among the participants seemed to be strongly gender-based. Saudi males were found to consume alcohol more than females did. This finding is consistent with the culture in Saudi Arabia. Additionally, the participants were conscious of drinking, yet few of them reported that they drank a lot. Most participants consumed alcohol quite frequently. However, the findings show a sense of awareness among the participants about the ill effects of alcohol. The most-reported health problems were blacking out and memory loss. It was also seen that a few patients who visited the ER had abnormal liver and pancreas function. In contrast, a significant number of participants from the community had liver and pancreas problems.

In line with the findings of other studies, the participants from the community reported that their initial encounters with alcohol had occurred early in life [12,18,19]. Two respondents shared that they had been consuming alcohol for 44 and 35 years, respectively. Moreover, the time spent drinking alcohol in a particular session was more than one hour for some participants. They also reported social, personal, and psychological effects of alcohol and deteriorations in their general health and mental health. For instance, patients arrived at the ER showing symptoms such as altered levels of consciousness, dizziness, vomiting, confusion, and jaundice. The third group of patients did not require hospital admission. They were discharged from the ER after initial treatment and returned home following a period of observation to ensure that they were well. Overall, many people in all three groups were found to drink alcohol and have associated health-related problems.

The findings of the study are consistent with another study that reported that approximately 10.6% of patients in Najran, Saudi Arabia were under the influence of heavy alcohol intake. However, unlike the present study’s results, in which one death was reported, the prior study demonstrated good patient outcomes [20]. A study conducted at Al-Amal Hospital in Jeddah noted that 16.1% of patients were consuming and abusing alcohol [21]. Another study reported that alcohol had become one of the most frequently consumed substances by Saudis and that psychosocial stress and peer pressure were the major causes of alcohol abuse [22].

Alcohol use is regarded as a major risk factor for the development of non-communicable diseases all over the world. The excessive consumption of alcohol causes millions of deaths annually and is the third most important risk factor for morbidity and disability [23]. A decline in alcohol use is related to a reduced risk of developing cardiac diseases. Nevertheless, despite the prohibition of drinking alcohol in Saudi Arabia, the current study revealed that most participants consumed alcohol and that its toxicity was prevalent in all three groups.

### Limitations

Since this was a retrospective study, it had limitations. First, some cases could not be included in the study due to their unclear data. Moreover, no information was available about patient outcomes following discharge from the hospital, for both Group 1 participants and for the interviewed community members from Group 2. During the study period, patients may have received healthcare at primary care centers and been discharged without referral to the central hospital. This means that it is likely that relevant cases were not included in this study. Moreover, no data were available regarding the outcomes (including deaths) after hospital discharge. Saudi males were found to consume alcohol more often than females, which could be due to the culture in Saudi Arabia, as in Group 1, or some females may not have responded to the questionnaire or interview in Group 2, which is also culture-related.

## 5. Conclusions

Alcohol consumption, together with intoxication, was recorded in the current study, although producing, selling, and drinking alcohol is a punishable crime in Saudi Arabia. Further, social influences and stress contributed to the initiation of alcohol use. Despite the ban, alcohol can be smuggled into Saudi Arabia and distributed via the black market, or it may be illegally homemade, which further prevents authorities from effectively intervening. Moreover, adulterated ethanol use can lead to methanol poisoning and death. Therefore, it can be stated that alcohol abuse is still a major threat to the Saudi population. Such evidence can be used by authorities and health policymakers to formulate plans for the control and prevention of alcohol abuse.

## Figures and Tables

**Figure 1 ijerph-19-15291-f001:**
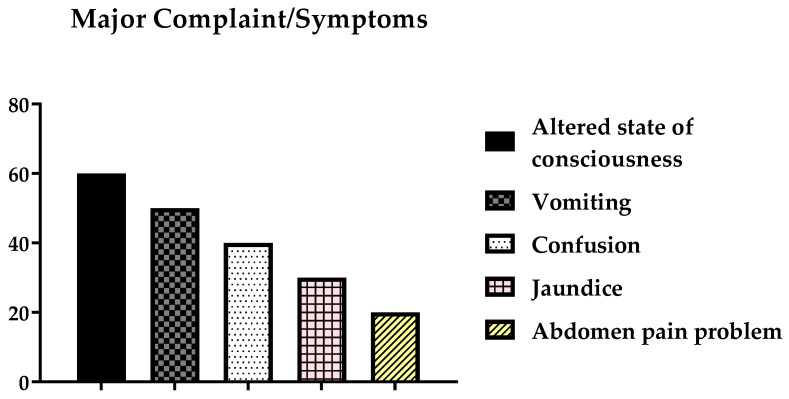
Distribution of the major complaints among the participants due to alcohol consumption.

**Figure 2 ijerph-19-15291-f002:**
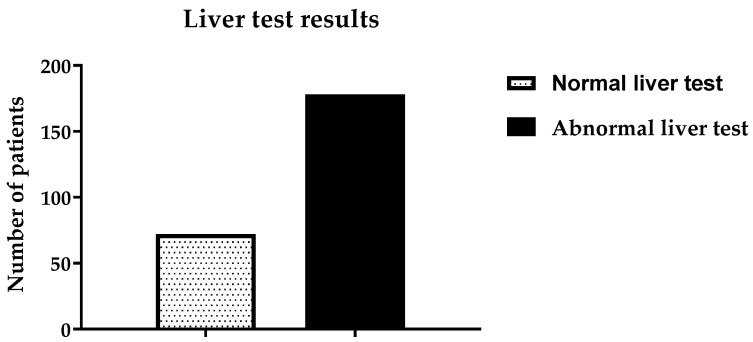
The liver function test results of all the participants.

**Figure 3 ijerph-19-15291-f003:**
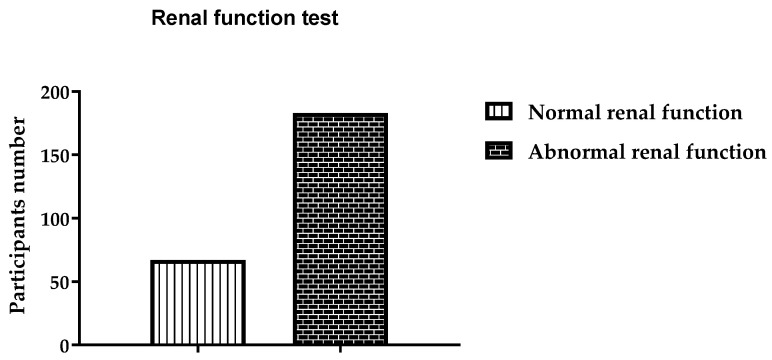
Renal function test results of all participants.

**Figure 4 ijerph-19-15291-f004:**
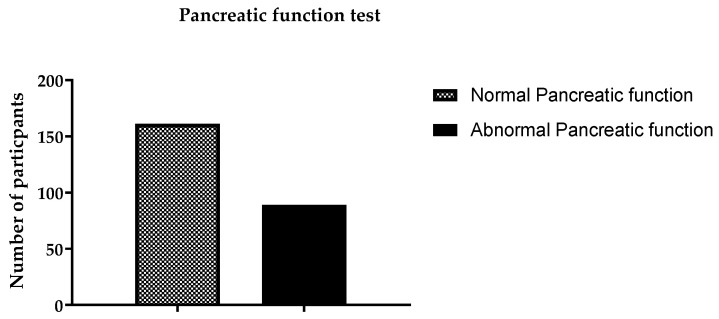
Pancreatic function test results of all participants.

**Figure 5 ijerph-19-15291-f005:**
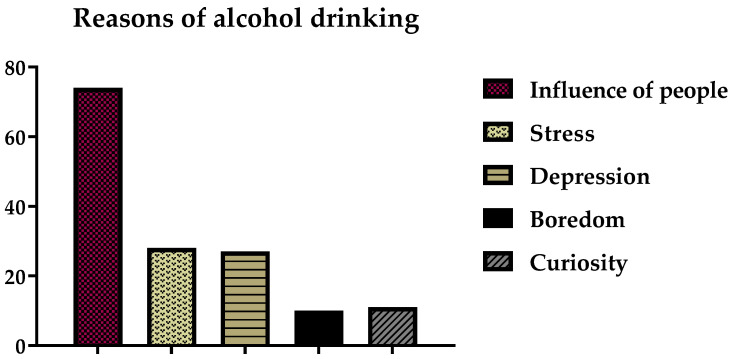
Reasons for alcohol drinking.

**Figure 6 ijerph-19-15291-f006:**
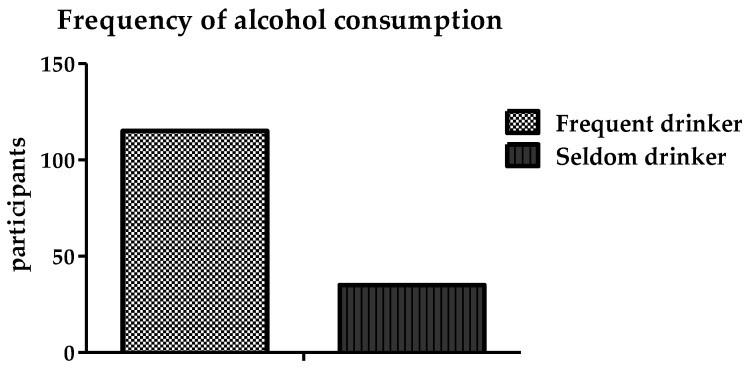
Frequency of alcohol consumption among the participants.

**Figure 7 ijerph-19-15291-f007:**
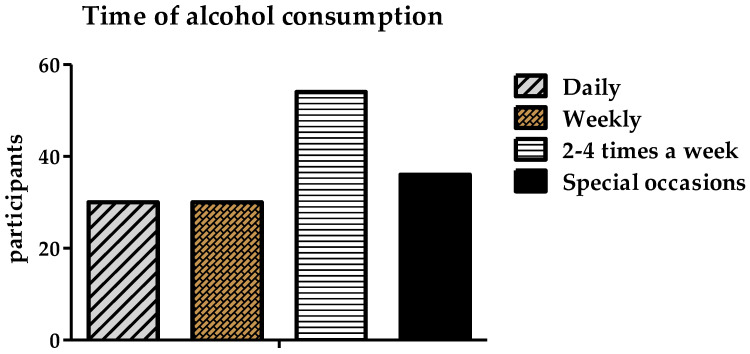
Frequency of alcohol consumption by the participants.

**Figure 8 ijerph-19-15291-f008:**
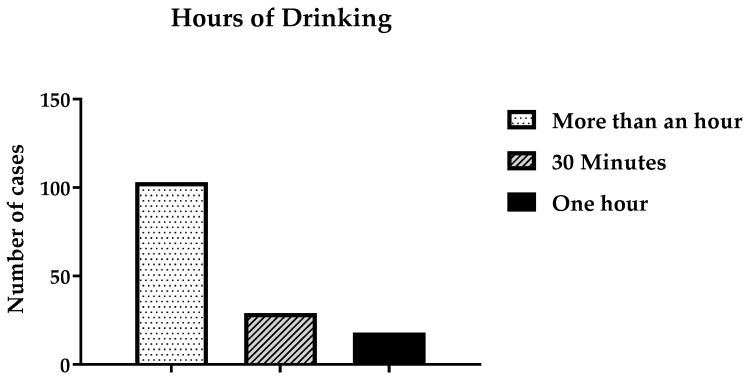
Time spent consuming alcohol.

**Figure 9 ijerph-19-15291-f009:**
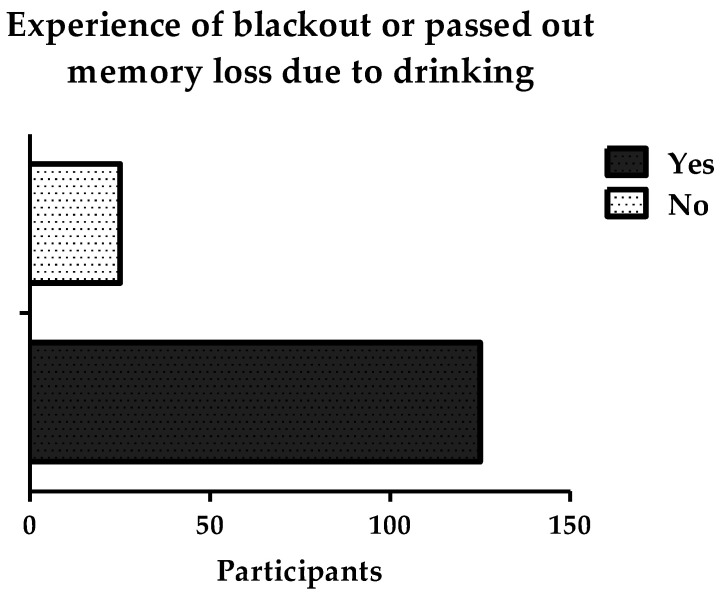
Participants’ experiences of blacking out, passing out, or memory loss because of alcohol consumption.

**Figure 10 ijerph-19-15291-f010:**
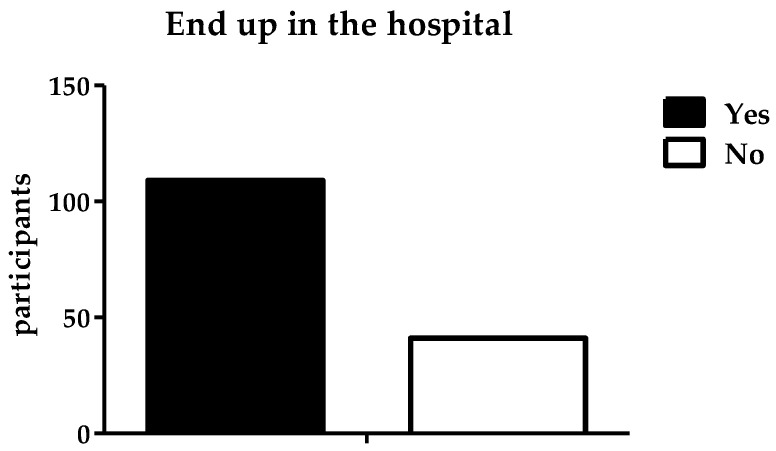
Number of participants who ended up in hospital.

**Figure 11 ijerph-19-15291-f011:**
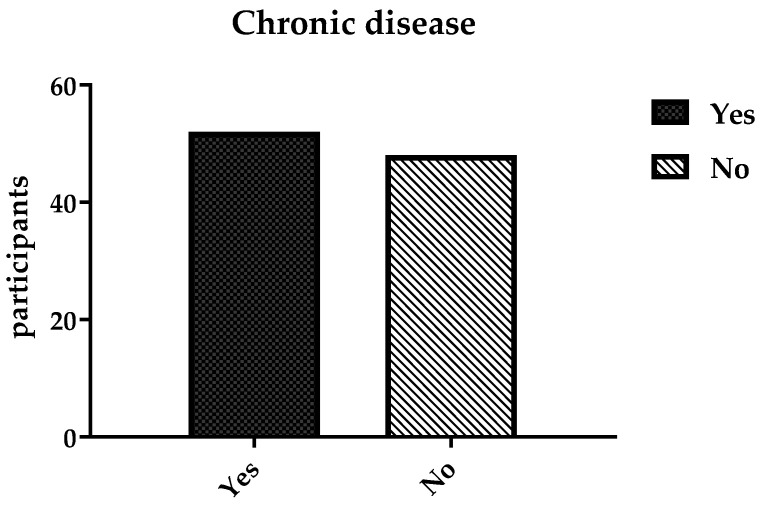
Number of chronic diseases among participants.

**Table 1 ijerph-19-15291-t001:** Demographic profile of Group 1.

Gender	
Male	244
Female	6
**Mean age in years**	28 (±9.2)
**No. of participants by age at occurrence of alcohol poisoning (years)**	
19–30	167
31–59	81
60 to 75	2
**Nationality**	
Saudi	241
Indian	8
Non-Saudi	1
**Marital status**	
Single	193
Married	55

**Table 2 ijerph-19-15291-t002:** Demographic profile of Group 2.

Gender	
Male	128
Female	22
Mean age in years	31(±10)
**No. of participants by age at start of alcohol consumption (years)**	
19 to 30	44
31 to 59	100
60 to 75	6
**Education**	
Elementary	1
High school	59
Intermediate	8
Bachelor	77
Postgraduate	5
**Nationality**	Saudi
**Marital status**	
Single	120
Married	30

## Data Availability

The data that support the findings of this study are openly available from the corresponding author upon reasonable request.

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
