# Peer review of "Reported Cases of Alcohol Consumption and Poisoning for the Years 2015 to 2022 in Hail, Saudi Arabia"

_ijerph, 2022, doi:10.3390/ijerph192215291_

Round 1

Reviewer 1 Report

I have read this paper with high interest and I think it could contribute to enhance knowledges in scientific research on drinking, but there are some part that need to be improved before to accept it for the pubblication.

-First of all the research regard a time between june 2015-june 2022. In this range a covid pandemic occurred and probably it affected the people's life increasing the time spent at home, levels of anxiety and depression that could be have influenced the alcohol consumption. So, I think that the authors need to take into consideration this variable also if they didn't misured it.

-The age range is very wide, 17-75. The groups are divided not respect the age, but the motivation that lead people to drink are very different during adolescence ora during adulthood, so I think that is important to take into consideration the possibility to divide the groups for age.

-The authors declare that the study is a qualitative research but there is a lack in the methodology. They refers to an interview but it isn' t presented the interview text, just as there isn't information about the data  analysis. Also in the tables there are lacks about DS of age. 

Reviewer 2 Report

The manuscript consists of total 14 pages, including 1 empty page, 11 figures, 2 tables and the list of total 23 literature references. The manuscript presents the original results of a study on alcohol abuse in one of the countries where alcohol use is prohibited. As such, it is an interesting topic fitting into the scope of works published in the Journal. The English language quality is acceptable in the major part of the text but it still could benefit from a correction by a native speaker experienced in scientific writing. Anyway, the text seems to be an advanced draft rather than the final version as there is page 2 which is blank, the group 3 results text presentation is absent and a working comment is present (line 259 page 10), as well as it is in the case of the discussion and limitations (line 326 page 12). As such, the text shall be rejected in its current form but the Authors may be encouraged to re-submit the finished, reworked, final form of the article.

The Abstract mirrors the major findings presented in the main text.

The Introduction provides enough background information to justify the study and provide context of the researched problem.

The Materials and methods section is divided into sub-sections for clarity. However, the reason behind forming the 3 groups needs to be justified here; as I understand, the Authors intended to provide the characteristics of: the severity level of alcohol abuse health detrimental results in the group 1, and the characteristics of the prevalent drinking patterns in the group 2; the group 3 results were not provided yet and thus the reason for investigating this group is unclear.

In case of the Results section, the results were collected and presented in quite subjective way, in particular the Authors did not provide the precise laboratory parameters but only their own interpretations of these parameters ("normal/abnormal", "frequent/seldom" etc.) that do not have any precise definitions; the Authors also decided against using any of the available widely accepted scales for alcohol dependency like Alcohol Dependence Scale (ADS), AUDIT questionnaire etc. which would allow to objectively compare the Authors' results against other published works; also the demographic profile data were not cross-analyzed with the other evaluated parameters.

The Figures shall follow a unified graphical convention and have number labels; current differences do not have any visible justification and make a chaotic impression.

The Conclusions are too general and do not refer to the Authors' concrete findings.

The literature references are numerous and recent enough, relevant to the raised problem.

Round 2

Reviewer 1 Report

Dear authors, I appreciated your work in improve the paper. I think that it is necessary only a last refine about the gender difference.

As your sample is very unbalanced by geneder, it is necessary to address this condition in your paper, and pay attention to it in discussion (line 277-279) and limitation section. The sentence in line 277-278 need to be reformulate on base on your sample-condition.

Reviewer 2 Report

Highly unfortunately, although the Authors stated that They have introduced the suggested changes to the text, in fact the merit of the previously listed problems stays the same in the new version, with only limited improvements.

The methodology that had been questioned previously has been better explained but not improved - which is reasonable since the study had been already completed. In the line 86 there apparently shall be "400" instead of "450".

The text fragments yellow highlighted by the Authors as altered stay in fact in significant portions the same as earlier.

The chaotic graphical presentations stay not unified.

The conclusions are too general and at the same too strong, not adequately supported by the presented study findings.
